# Linear, High Dynamic Range Isolated Skin Resistance Transducer Circuit for Neurophysiological Research in Individuals after Spinal Cord Injury

**Martin Vítězník** [1,*] ⓘ, **Tomáš Veselý** [1] ⓘ, **Radim Kliment** [1] ⓘ, **Pavel Smrčka** [1] **and Jiri Kriz** [2] ⓘ

1  Department of Information and Communication Technologies in Medicine, Faculty of Biomedical Engineering, CTU in Prague, 128 00 Prague, Czech Republic; tomas.vesely@fbmi.cvut.cz (T.V.); radim.kliment@fbmi.cvut.cz (R.K.); smrcka@fbmi.cvut.cz (P.S.)
2  Spinal Cord Unit, Motol University Hospital, 150 06 Prague, Czech Republic; jiri.kriz@fnmotol.cz
*  Correspondence: martin.viteznik@fbmi.cvut.cz; Tel.: +420-224-968-574

**Abstract:** The quantification of skin resistance in individuals after spinal cord injury for the purpose of neurophysiological research is difficult, mainly as a consequence of decreased activity of sweat glands in the injured human organism. In this original work, we propose a custom electrical skin resistance transducer, featuring extremely low patient auxiliary current, linear response and high dynamic range. After the design and fabrication of the prototype device, we conducted preliminary benchmark tests. We found that our prototype transducer was able to linearly report a broad range of resistance presented to its input terminals, which is not usually found in skin resistance research instrumentation. The basic design idea is viable and, following further research, an improved version of presented prototype device may be used for the purpose of neurophysiological research in individuals after spinal cord injury.

**Keywords:** galvanic skin response; electrical skin resistance; transducer; spinal cord injury

## 1. Introduction

Electrodermal activity, also known under various other terms including "galvanic skin response", "skin conductance level" or "skin resistance", has been routinely evaluated for decades to assess the reaction of an individual's autonomic nervous system to various psychical and physical stimuli [1]. Change in the galvanic skin response is one of the set of biological signals routinely recorded in a polygraph procedure during criminal interrogation (so-called "lie detector"). This is, among the general public, probably the most well-known application of measurement of electrodermal activity [2–4]. Although based on solid foundations, the outcome of such an application is nowadays discussed because certain (voluntary) methods exist, which are able to mask (involuntary) the reaction of the subject and render the whole examination questionable [5–7].

Despite the fact that polygraph devices are depicted as widely utilized in popular culture, modern forensic science is in search of different methods that could potentially quantify a suspect's level of mental stress and hence identify any unusual behaviors of the interrogated person when faced with a series of specific, custom-tailored questions. The level of electrodermal activity or, particularly, skin conductance level is, however, still an important parameter evaluated, e.g., in experimental human psychology research [8–10]. Observed changes in those parameters are in close relation to the activity of different parts of the human autonomic nervous system: whenever a certain stress stimulus arises, a subsequent reaction of the sympathetic nervous system leads to the increased activity of the sweat glands located under the surface of skin. Two important characteristics of this process need to be considered. First, such phenomena are out of the voluntary control of the subject's mind; second, sweat produced in eccrine sweat glands and brought to the skin

surface by sweat pores affects electrical conductance of specific surface areas of the body. This effect is supported by the content of sodium chloride in human sweat, which acts as an exquisite electrolyte.

In associated research, our colleagues focused on the evaluation of the functions of the autonomic nervous system in individuals affected by spinal cord injury. This condition is the result of mechanical trauma to the spine, and its most common causes include traffic accidents, sport injuries or falls. Since the trauma cuts specific neural paths in the human body, among other difficulties, injured individuals exhibit a considerable decline in their sweat gland activity. As a consequence, the electrical skin resistance of the individuals in question is noticeably high, depending on the severity of the injury, and is reported to be in the range of 5–10 MΩ [11]. In non-injured individuals, the common value is well under 1 MΩ [12]. This is the reason why it is not possible to incorporate usual galvanic skin response measurement devices in such research and therefore also serves as a motivation for the development of an experimental skin resistance transducer.

It must be emphasized that the variations of skin resistance and electrodermal activity in general have been routinely measured and assessed for decades. This broad topic is extensively described elsewhere [13]. Traditionally, in psychology, medicine and related fields of science, skin resistance ($R$) is instead reported as skin conductance ($G$) [14]. Both quantities are inversely proportional ($G = 1/R$). Observations are usually expressed as *SCL* (skin conductance level) and *SCR* (skin conductance response). This follows the electrical model of electrodermal activity with resistance of sweat ducts connected in parallel, as extensively described in [13] (pp. 71–73). The typical parameters of a recorded signal and recommendations for instrumentation performance are summarized in [13] (pp. 151–157). However, since this report presents an evaluation of the experimental instrument based on electrical engineering approach, we are reporting our results in terms of electrical resistance.

## 2. Materials and Methods

### 2.1. Overview

In response to the requirements that arose within a broader research project, we developed the following procedure, summarized as a list below. Each step is thoroughly described later in the text.

- Identification and analysis of requirements;
- Synthesis of custom transducer circuit and selection of appropriate components;
- Fabrication of a prototype device;
- Assessment and evaluation of performance in controlled laboratory environment (in vitro).

### 2.2. Identification and Analysis of Requirements

Within the scope of ongoing research related to the function of the autonomic nervous system in individuals after spinal cord injury, our task was to propose a method and device for the continuous monitoring of electrical skin resistance in such subjects during specific clinical examination. Several limitations of state-of-the-art devices exist, which are in close relation to special conditions in individuals after spinal cord injury.

The key requested feature, not found in commercially off the shelf available devices, was an ability to handle extremely high levels of skin resistance. One of the serious consequences of damage to the human spine is a reduction in activity of eccrine sweat glands. This does not only limit the evaporative thermoregulation of the body, but also significantly affects the levels of electrical skin conductivity. Based on previous research [11], the request for a useful range was targeted at 10 MΩ. Nevertheless, neither an ability to report low resistance values (<100 kΩ), nor high resolution of the transducer was needed within the associated experiments. In previous research [11], electrical skin resistance was reported to follow certain neurological stimuli with the delay in the order of seconds and slew rate of about 50–100 kΩ/s.

Furthermore, the output of the signal was requested to be in real time, in terms of analogue voltage, directly proportional to presented skin resistance across input terminals. This was required with respect to compatibility with the signal recording device routinely used by our research partners, and the conversion factor was agreed to be 1 V per 1 MΩ.

Finally, the method and device in question was required to comply with ISO/IEC 60601-1 (international standard applicable for electric medical devices) [15] in terms of highest applicable auxiliary patient current.

### 2.3. Synthesis of Custom Transducer Circuit and Selection of Appropriate Components

#### 2.3.1. General Design Approach

With respect to the requirements set, we proposed an idea of transducer circuit, which is depicted in the Figure 1. The circuit is based on a reference current constant source feeding the resistance under test. As a consequence of (constant) electrical current being forced through the unknown resistance, a voltage drop establishes over the terminals. This signal is then coupled with respect to its high source impedance and processed in low pass filter. The filtered signal is then fed into an isolation amplifier and brought to the output terminals.

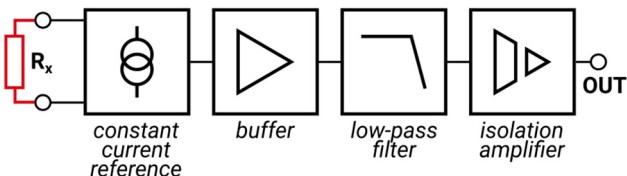

**Figure 1.** Block diagram of principal idea of custom transducer circuit.

With regard to Ohm's law, the observed voltage is proportional to the reference current flowing through an unknown resistance (expected to be surface resistance of skin in the future application of our device). In order to achieve the requested conversion factor (1 V per 1 MΩ), the current source shall deliver the current I = 1 μA. The preamplifier acts solely as a unity gain buffer.

#### 2.3.2. Actual Design of the Prototype Device

Following an approach depicted in idea design, we synthesized the circuit of prototype transducer device, which is shown in detail in Figure 2. With reference to the constant current source, integrated dual 100 μA source/sink REF200 (Texas Instruments, Dallas, TX, USA) was used. By coupling together with precision a low-distortion, extremely low input bias current operational amplifier OPA602 (BurrBrown/Texas Instruments, Dallas, TX, USA), as recommended in [16], and an appropriate resistor network (YAGEO, New Taipei City, Taiwan), we were able to not only obtain constant current source of 1 μA, but also measure electrical voltage at the output of the operational amplifier, which is directly related to the voltage present at the non-inverting input and thus reflects the unknown resistance.

The output voltage of the previous stage was then coupled through a unity gain voltage buffer into an active low-pass second-order filter (passband 5 Hz). Both functional blocks were constructed using a precision, rail-to-rail input output, low offset voltage operational amplifier OPA2197 (Texas Instruments, Dallas, TX, USA).

The signal at the output of previous (active filter) stage was then fed into a highly linear, unity gain precision isolation amplifier ISO124P (Texas Instruments, Dallas, TX, USA) and from the output of this amplifier to the output terminals.

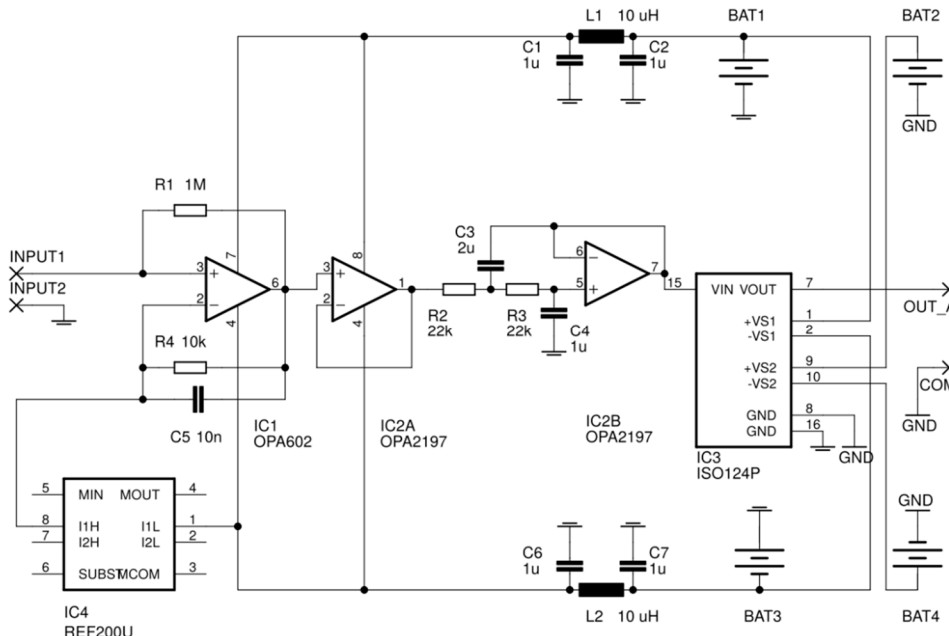

**Figure 2.** Schematic diagram of the circuit of prototype transducer device.

## 2.4. Fabrication of Prototype Device

We developed a custom, double-sided printed circuit board for the circuit using appropriate software tools (EAGLE, Autodesk Inc., San Rafael, CA, USA) and fabricated the board in cooperation with a local manufacturer using FR4 glass-epoxy laminate DURAVER-E-Cu quality 104 (Isola GmbH, Düren, Germany). The printed circuit board was then manually populated in our laboratory using RoHS certified Sn95.5Ag3.8Cu0.7 solder (Stannol GmbH & Co. KG, Wuppertal, Germany) and ST50 temperature-controlled soldering station (PACE Inc., Vass, NC, USA).

Keeping in mind that the circuit was designed to work with extremely low electrical currents and high resistances, in order to minimize the effect of air moisture on the circuit, we applied special acrylate colorless conformal coating PLASTIK70 (CRC Industries Europe BV, Zele, Belgium) to all surfaces of the assembled printed circuit board including all components, as well as to free unassembled component pads. Before applying conformal coating, the whole assembled printed circuit board was thoroughly cleaned multiple times using paper wipes and isopropyl alcohol (Lach-Ner s.r.o., Neratovice, Czech Republic) to remove possible residuals of flux and other contamination, which could adversely affect its function.

We decided to use 14 V lithium polymer rechargeable battery cells as a primary mean of powering the prototype device (Hextronik Limited, Kwun Tong, Hong Kong).

## 2.5. Assessment and Evaluation of Performance in Controlled Laboratory Environment (In Vitro)

In order to estimate the transfer function of the experimental transducer in question and evaluate its performance, we applied various resistances to the input of the device by means of different discrete resistors (Vishay Intertechnology Inc., Malvern, PA, USA). All discrete measurements were obtained using Keysight 34461A, a precision $6\frac{1}{2}$ digit digital multimeter featuring valid calibration (Keysight Technologies, Santa Rosa, CA, USA). We used averaging over $n = 100$ samples for all measurements with the exception in case of repeatability assessment.

Furthermore, we set up a model circuit similar to the intended application including two self-adhesive Ag/AgCl electrodes WhiteSensor WS (Ambu A/S, Ballerup, Denmark), which are routinely used to interface biosignals from the human body, in order to assess the slew rate of the output of the prototype device (see Figure 3). The setup comprised two stainless steel plates with attached self-adhesive electrodes connected to a shuntable

resistor network. Temporal characteristics were recorded using a digital signal oscilloscope DSO6054A (Agilent Technologies Inc., Santa Clara, CA, USA).

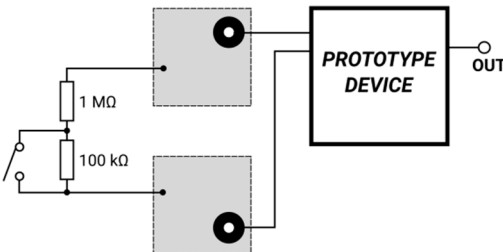

**Figure 3.** Experimental setup including Ag/AgCl electrode interface (self-adhesive electrodes attached to stainless steel plates) used in assessment of slew rate of the prototype device.

Characterization of repeatability of the custom device was performed by measuring constant resistance multiple times ($n$ = 11) and evaluating the standard deviation of the acquired data. We also estimated noise amplitude of the output while connected to the constant resistance (1 M$\Omega$).

Standard laboratory test leads and probes were used to connect the prototype device to the test equipment. All measurements were performed under normal laboratory conditions.

*2.6. Overall Evaluation*

For the purpose of systematically identifying a difference in measurements between the gold standard instrument (precision digital multimeter in our case) and our prototype transducer, we settled on taking advantage of the following methodology, extensively used in biomedicine [17]. Obtained values were used partly to construct a Bland–Altman plot of agreement between methods, partly to construct an xy scatterplot that allows us to compare the measurements to the line of equality.

The resulting plots were constructed using software package R (R Core Team, Vienna, Austria) [18] in conjunction with ggplot package [19]. Our interest was focused on two main questions: first, whether the proposed method would be suitable for a transducer with high dynamic range and second, if a sufficient resolution could be obtained despite the extremely high dynamic range of the prototype transducer. The aim was set for a resolution of 10 k$\Omega$: when baseline skin resistance is R = 1 M$\Omega$ (G = 1 $\mu$S), this value represents the minimal expected electrodermal response of 0.01 $\mu$S, as described in [14].

**3. Results**

Based on experimental data and the provided methodology, our results were entered into a Bland–Altman (mean-difference) plot (see Figure 4) and xy scatterplot along with a line of equality (see Figure 5). Full original raw data are presented in Table 1.

As we can clearly see from the plot on Figure 4, the experimental device shows certain measurement offset. Mean difference was computed at −1.6 k$\Omega$ with 95% limits of agreement (lower: −6.6 k$\Omega$; upper: 3.4 k$\Omega$). This offset is not of constant value but is rather linearly dependent with respect to the resistance value presented to the input terminals. For all measurements, the observed difference is still within 95% limits of agreement.

The xy scatterplot in Figure 5 indicates that there were no outliers nor seriously incorrect values measured throughout the experiment. Furthermore, as can be seen, the experimental device exhibits linear transducer characteristics across the whole specified range.

By comparing the raw measured values, we observed a steady offset of the experimental prototype transducer device, which is, in a more illustrative manner, also evident in Figure 4.

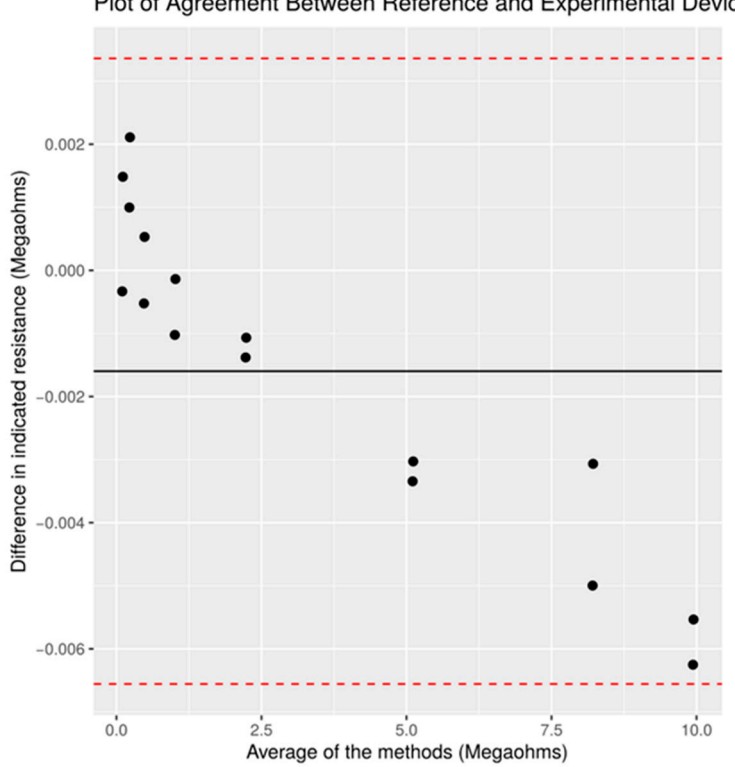

**Figure 4.** Bland–Altman plot of agreement between reference device and experimental transducer device.

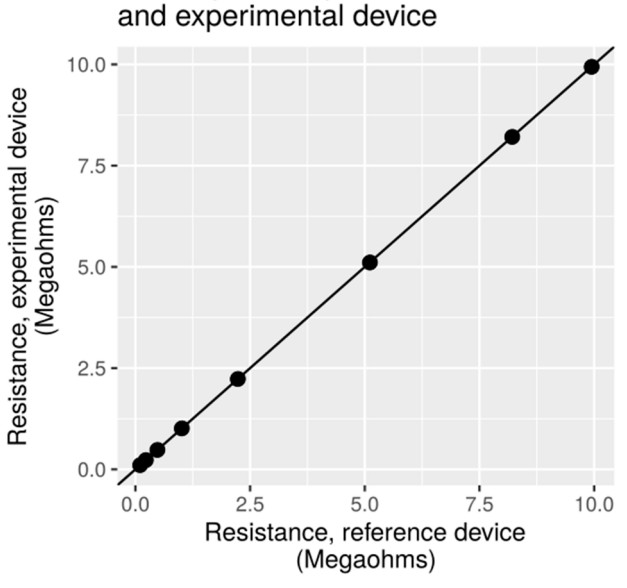

**Figure 5.** Scatterplot of indicated values (reference vs. experimental device) featuring line of equality.

**Table 1.** Full original data showing reading of series of test resistors when measured by reference device compared to value indicated by prototype transducer; averaged values (*n* = 100), rounded to 6 decimal places.

| Reference Device Reading (MΩ) | Prototype Transducer Reading (MΩ) |
|---|---|
| 0.099792 | 0.099459 |
| 0.109772 | 0.111255 |
| 0.221909 | 0.222905 |
| 0.231899 | 0.234009 |
| 0.474827 | 0.474303 |
| 0.484803 | 0.485333 |
| 1.007170 | 1.006146 |
| 1.017144 | 1.017006 |
| 2.228005 | 2.226624 |
| 2.238135 | 2.237067 |
| 5.107629 | 5.104285 |
| 5.117086 | 5.114057 |
| 8.210684 | 8.205687 |
| 8.218246 | 8.215179 |
| 9.942075 | 9.935823 |
| 9.950774 | 9.945239 |

When estimating the step response of the prototype transducer, for instant change of resistance ΔR = 100 kΩ, the observed time constant was 33 ms. The recorded waveform is presented in Figure 6.

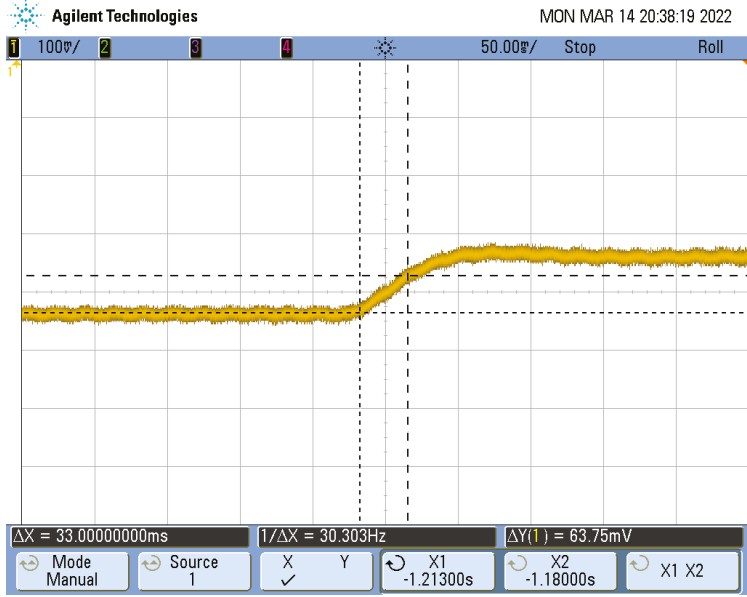

**Figure 6.** Step response of experimental transducer setup (see Figure 3) recorded by digital oscilloscope. Cursors represent measurement of time constant (33 ms), which leads to cutoff frequency of 4.8 Hz.

We performed a repeatability characterization measurement by measuring two different resistors (nominal resistance: 1 MΩ and 5.1 MΩ) with the experimental device multiple times (*n* = 11) and evaluating the deviation of the data. Both datasets were checked for normality using the Shapiro–Wilk test ($p = 0.45$ and $p = 0.59$, respectively). The results are presented in Table 2.

**Table 2.** Repeatability characterization of the prototype device ($n$ = 11).

| Nominal Resistance | Measured Mean Value | SD |
|---|---|---|
| 1 MΩ | 1.007 MΩ | 0.23 kΩ |
| 5.1 MΩ | 5.102 MΩ | 0.19 kΩ |

In order to assess the noise level of the device, we connected a fixed resistor (1 MΩ) to the input and measured amplitude (peak–peak) by an AC-coupled oscilloscope (see Figure 7, channel 1). The mean peak–peak noise amplitude was measured at 2.0 mV ± 0.07 mV ($n$ = 1032). Data from channel 2 on the figure represented shorted probe and are included for comparation.

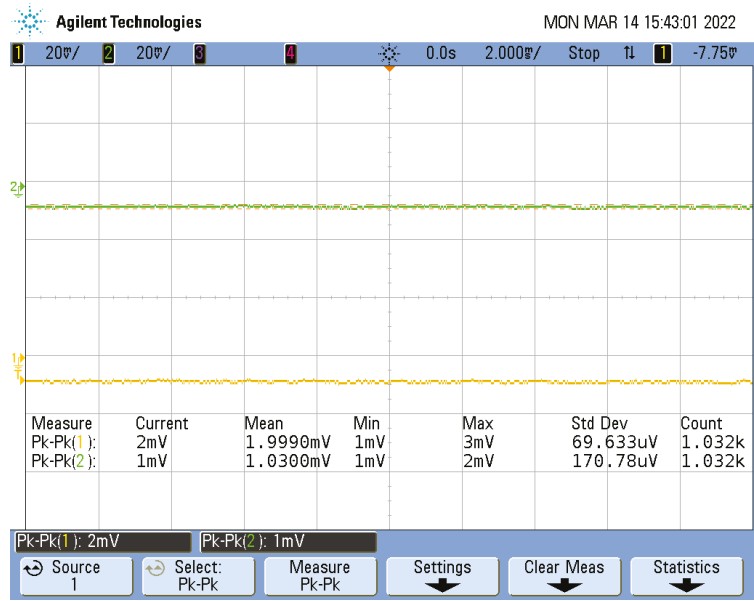

**Figure 7.** Noise at the output of the experimental device (with 1 MΩ resistor at the test terminals). Channel 1 (yellow): actual output of the device. Channel 2 (green): shorted probe (for comparison purposes).

## 4. Discussion

After a careful evaluation of our original experimental research data, it is possible to clearly demonstrate the certain ability of the studied experimental transducer setup to report a broad range of resistance presented to its input terminals. Such range and ability to linearly report resistance values up to 10 MΩ are not commonly found in commercially available devices intended for performing skin resistance measurements in human subjects. This is the key characteristic for the planned future use of the custom transducer in question.

However, the performance of the experimental device was found to be limited to some extent. As can be deduced from the plot of agreement between the reference device and experimental transducer device in Figure 4, the observed values of difference follow a linear trend. Mean differences between instruments were found to be −1.6 kΩ with 95% limits of agreement (lower −6.6 kΩ; upper 3.4 kΩ). Additionally, when comparing relative values of difference, such a value is in proportion with the reported output value, which is notably large.

When testing the resolution of the system, by increasing connected resistance by 10 kΩ, we were clearly able to detect such change proportionally in the output signal.

It is important to point out that, while performing the test measurement using the largest test resistor within the set (10 MΩ), the isolation amplifier present in the signal chain (feeding the output terminal) worked at its amplitude limits (specified ±10 V). This fact may have also introduced a certain level of inaccuracy into the signal. Apart from this, another

limitation exists in the range of lower resistance values (<100 kΩ). Since we intentionally used an extremely small electric current (1 μA) for excitation, the observed voltage lies in the range of microvolts. This low voltage is somewhat inconvenient to process and handle, also with respect to the usual electrically noisy environment of a standard laboratory or hospital room.

With regard to the set of international standards for medical devices (IEC 60601) and its limitation for an auxiliary electrical current (applied to a patient), our experimental transducer notably fulfils the criteria listed in the standard.

Future research plans for our team include the development of an improved version of the experimental transducer. Our approach will include the selection of tight tolerance components, as well as the acquisition of more elaborate laboratory instrumentation, notably, a precision ohmmeter with sufficient range.

We found the slew rate of our experimental setup (response to step change in presented resistance, connection using standard biosignal electrodes) to be approximately 1.9 MΩ/s. This fulfils the needs of our colleagues, since the observed rate of change of electric skin resistance in target patients was found to be roughly 20–100 kΩ/s. The measured slew rate was in agreement with the 5 Hz cutoff frequency of the implemented low-pass filter.

The repeatability of measurement was tested by multiple measurements of two different resistors (1 MΩ and 5.1 MΩ). The observed standard deviations (see Table 2) showed that the device gives repeatable readings both for lower and higher resistance values.

The device exhibited a noise on its output, with mean amplitude (peak–peak) of 2.0 mV ± 0.07 mV. This amplitude of noise effectively meant noise in converted value being approximately 2 kΩ.

It is our assumption that, after incorporating the abovementioned minor improvements to our setup, we will be able to conduct a standard calibration procedure of the prototype device and accomplish the specified goals with certainty.

## 5. Conclusions

In this work, we presented an experimental linear, high dynamic range isolated skin resistance transducer, which was specifically designed to support neurophysiological research in individuals after spinal cord injury. The need for such a device arose from specific conditions found in such individuals, most notably decreased sweat gland activity, which rendered standard devices unusable. We found and demonstrated that our original design approach is viable: it is possible using the circuit to sense resistance over the range 0.1–10 MΩ, with a resolution of 10 kΩ, fair reproducibility (SD = 0.2 kΩ) and an equivalent noise amplitude of 2 kΩ. Resistance under the test is galvanically isolated from the rest of the circuit and signal recorder. Following a little further research, an improved version of the presented prototype device may be used for the designed purpose.

**Author Contributions:** Conceptualization, J.K. and P.S.; methodology, M.V. and P.S.; validation, T.V. and R.K.; formal analysis, R.K. and P.S.; investigation, M.V.; resources, J.K. and P.S.; data curation, M.V.; writing—original draft preparation, M.V.; writing—review and editing, M.V.; visualization, T.V.; supervision, M.V.; project administration, P.S.; funding acquisition, M.V. All authors have read and agreed to the published version of the manuscript.

**Funding:** This research was funded by the Grant Agency of Czech Technical University in Prague, grant No. SGS21/183/OHK4/3T/17. The APC was funded by Czech Technical University in Prague, Faculty of Biomedical Engineering, Department of Information and Communication Technology in Medicine.

**Data Availability Statement:** Not applicable.

**Acknowledgments:** The authors would like to express thanks to their home institutions and funding provider. Exceptional acknowledgement belongs to Jan Dudák (Institute of Experimental and Applied Physics, Czech Technical University in Prague) for a loan of precision digital multimeter.

**Conflicts of Interest:** The authors declare no conflict of interest. The corresponding author receives funding by his home institution, which also endorses publication of the results.

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
