# Peer review of "Linear, High Dynamic Range Isolated Skin Resistance Transducer Circuit for Neurophysiological Research in Individuals after Spinal Cord Injury"

_electronics, doi:10.3390/electronics11071121_

Round 1

Reviewer 1 Report

Authors showed quantification of skin resistance using custom made electrical skin resistance transducer with high dynamic range. The manuscript seems to be clear and good for brief report. However, there are some addition analysis data with following description should be needed even though this is Brief report because developed electronics for the transducer is well described. Provided information for the tools and measurement setup is well described. Therefore, authors could be revise the manuscript accordingly so I can recommend the manuscript.

1. Authors mentioned that transducer is developed. However, there is electronics. It is better to change the title.
2. In Figure 5, fonts are not clear.
3. In Figure 6, authors need to mark the important data.
4. Authors must provide some additional information in Figure 6 such as delay time.
5. Authors had better provide some important data in the conclusion section.
6. In Figure 1, authors had better show what are blocks.
7. Please provide gain value of the preamplifier (voltage amplifier).

Author Response

Dear Reviewer,

thank you for your helpful comments and review. Please find my response below.

  • We updated the title according to your suggestion.
  • As recommended by second reviewer, we have re-done the measurements and thus Figures 5 and 6 are updated. The fonts look clear now, I hope there are no issues with MS Word compatibility.
  • Regarding Figure 6 (step response), we provided detailed information in the figure caption.
  • Important data were added in the conclusion section.
  • Figure 1 was redrawn to include description of the blocks.
  • Information on gain value of the amplifier (unity gain, G=1) was added to the text.

With regards
M. Viteznik

Reviewer 2 Report

Please refer to the attached file

Author Response

Dear Reviewer,

thank you for your helpful comments and review. Please find my response below.

  • We have re-done all the measurements using 6.5 digit precision multimeter.
  • Regarding the passband: thank you for pointing out this shortcoming and disorder in our materials. We corrected the schematic and also retested the response of the circuit.
  • We estimated noise amplitude of the circuit.
  • Characterization of repeatability was also added.
  • We added reference to appropriate literature as suggested, as well as paragraph related to expression of conductance/resistance values.

If needed, a file with raw measurement data is available at https://filesender.cesnet.cz/?s=download&token=2f4bb402-eeeb-4c7a-aa02-12eeb0374234 .

Best regards
M. Viteznik

Round 2

Reviewer 2 Report

I would like to thank the authors who have taken into consideration the recommendations given and have provided corrections and complementary information to improve the quality of the manuscript. Please see the attached file for complementary comments. A few changes might be necessary to give the paper the best possible quality.

Author Response

Dear Reviewer,

I would like to thank you for your helpful second review.

  • We corrected typos and got the manuscript proofread again.
  • Problems with format (disappeared figures etc.) probably originated when the editor went through accepting the changes. Please check PDF version of the document, if needed.
  • We reformulated the paragraph in the end of Introduction, as suggested in your review.

With regards
M. Viteznik